# Preparation of Solid Dispersion of Polygonum Cuspidatum Extract by Hot Melt Extrusion to Enhance Oral Bioavailability of Resveratrol

**DOI:** 10.3390/molecules28020737

**Published:** 2023-01-11

**Authors:** Wenling Fan, Jiali Wu, Meiqi Gao, Xiaotong Zhang, Wenjing Zhu

**Affiliations:** 1College of Pharmacy, Nanjing University of Chinese Medicine, Nanjing 210023, China; 2Jiangsu Provincial Technology Engineering Research Center of TCM Health Preservation, Nanjing 210023, China

**Keywords:** resveratrol, polygonum cuspidatum extract, hot-melt extrusion, solid dispersion, attendant substances

## Abstract

The aim of this study was to improve the solubility, bioavailability, and stability of resveratrol (RES-SD) Solid Dispersion in *Polygonum cuspidatum* extract (PCE) by hot melt extrusion (HME). In addition, the role of the auxiliary substances in PCE was also studied. The solid dispersion of *Polygonum cuspidatum* extract was prepared by hot-melt extrusion. The optimum formula was selected by single factor design and orthogonal test. The optimum formula was barrel temperature 140 °C, screw rotation speed 40 rpm/min, and the ratio of *Polygonum cuspidatum* extract to HPMCAS was 1:2. The dissolution test showed that PCE-SD increased the dissolution of RES from 46.75 ± 0.47% to 130.06 ± 0.12%. The pharmacokinetics curve of rats showed that PCE-SD increased AUC0-t of RES from 111,471.22 ± 11.4% to 160,458.968 ± 15.7%, indicating an approximately 1.44-fold increase in absorption. In addition, the rotation speed of PCE-SD screw is less than that of RES-SD screw. The bioavailability of PCE-SD was slightly better than that of RES-SD. PCE-SD is more hygroscopic than RES-SD. PCE-SD increased the solubility and oral bioavailability of RES. The auxiliary substances in Polygonum cuspidatum extract have influence on its preparation technology, stability, and bioavailability.

## 1. Introduction

Resveratrol (trans-3,5,40-trihydroxystilbene, RES) has unique biochemical and physiological effects, including activity against inflammation, several types of tumor, neurological disorders, cardiovascular diseases, diabetes, aging, and obesity [1,2,3,4]. However, the oral bioavailability of RES is extremely low, which leads tot the application of RES in human clinical practice being very limited [5,6], and the transformation research and clinical trials of RES have made little progress [7]. Therefore, improving the bioavailability of RES is the main challenge for its successful application in clinical and health promotion interventions. The poor oral bioavailability of RES is mainly due to its rapid metabolism and low water solubility [8,9]. Over the past years, various methods have been applied to improve the water solubility and bioavailability of RES, including the RES encapsulation in lipid nanocarriers or liposomes, emulsions, micelles, insertion into polymeric nanoparticles, solid dispersions, and nanocrystals [10,11,12,13,14,15,16,17].

In recent years, as plant extracts have been more and more widely used in the medical and health care fields in all over the world [18,19], a thought-provoking important issue has been placed in front of researchers: “the complex composition of extracts can cause differences in activity between pure substances and plant extracts.” For instance, a plant preparation prepared from the fruit of AMMI visagae, which has higher solubility than the pure furan chromone isolated from it. In view of this special nature of botanical drugs, Guo et al. [20] proposed that plant extracts should be divided into basic active substances and attendant substances. Among them, the basic active substances are completely or roughly related to the therapeutic properties of their preparations. However, there are few studies on the influence of accompanying substances on basic active substances.

Solid dispersion (SD) is a drug delivery system formed by converting poorly soluble active pharmaceutical ingredients into an amorphous state in a suitable carrier. Due to the high dispersion of insoluble drugs in SD, SD plays a unique role in the following aspects: improving the solubility and oral bioavailability of insoluble drugs [21], delaying the release time of drugs [22], controlling the release position of drugs in the body [23], and so on. Among the numerous preparation methods of SD, hot melt extrusion (HME) is a promising solvent-free continuous process and is suitable for pharmaceutical industrial production, which has attracted a great deal of attention in the pharmaceutical industry. Continuous extrusion processes are controlled by the hot melt extruder panel, and the extrudate could be pelletized, milled, mixed and 3D printed to dosage forms such as tablets, film, granules, capsules, and so on, which promotes the scalability of this technology. Several HME commercial products approved by the Food and Drug Administration (FDA), currently available in the market with different release targets, prove the potential and success of this technology [24,25,26,27].

At present, most commercial resveratrol is extracted from *Polygonum cuspidatum* [28]. In addition, in pH 6.8 phosphate buffer, the saturation solubility of RES in *Polygonum Cuspidatum* extract (PCE) is higher than pure RES in our previous experiments. Consequently, in this study, *Polygonum cuspidatum* extract was made into solid dispersion by hot melt extrusion technology. For one thing, PCE-SD can improve bioavailability of RES; for another, the effects of attendant substances in *Polygonum cuspidatum* extract on the preparation process, stability, and bioavailability of the solid dispersion were explored. The investigation of the biopharmacological and formulation effects of concomitant substances on pure active substances can provide a certain theoretical basis, data support, and scientific guidance for the separation and refinement of Chinese medicine extracts, and also has important academic value and practical significance for the reform of new dosage forms of Chinese medicine.

## 2. Results

### 2.1. Preparation of Polygonum Cuspidatum Extract (PCE)

The resveratrol contents of the three batches of *Polygonum cuspidatum* extracts were 50.15%, 50.43%, and 50.08%, the average content was 50.22%, and the RSD value was 0.37%. The preparation process was stable and feasible. The yield of resveratrol was 1.34%.

### 2.2. Results of Preliminary Qualitative Study on Main Attendant Substances in PCE

According to the physical and chemical characteristics of the chemical components contained in Polygonum cuspidatum [29], the following qualitative research was conducted, and the results are shown in Table 1.

### 2.3. Univariate Analysis of Solid Dispersion of Polygonum Cuspidatum Extract

#### 2.3.1. Type of Carrier

It can be seen from Figure 1 that on the DSC curve of the solid dispersion with EPO as the carrier, there is a less obvious crystal peak at about 230 °C, and it is preliminarily concluded that resveratrol crystals still exist in the extrusion. The DSC diagram of other extrudates has no obvious crystal peak, so the solubilization effect needs to be verified by dissolution experiment. It can be seen from Figure 2 that the dissolution effect of solid dispersion with EPO, PVP VA64, and soluplus as carriers is very poor. The solid dispersion of *Polygonum cuspidatum* extract prepared with hpmcas as carrier can significantly improve the dissolution of resveratrol. Therefore, hpmcas is selected as the carrier for preparing solid dispersion of *Polygonum cuspidatum* extract.

#### 2.3.2. The Type and Dosage of Plasticizer

It can be seen from Figure 3 and Figure 4 that the dissolution effect of solid dispersion with P188 is better than that with PEG6000, and better than that without PEG6000. The dissolution of solid dispersion without plasticizer has a great difference in the early stage. Considering the consistency of controlling the dissolution of solid dispersion and the increase in torque after reducing the extrusion temperature, plasticizer P188 is determined to be used to soften the material. In the experiment, the plasticizer dosage of 10% and 15% was investigated, but the extrudate was broken and could not be formed, so the dosage of P188 was 5%.

#### 2.3.3. Cooling Temperature

Figure 5 and Figure 6 shows that the dissolution effects of solid dispersions prepared by the three cooling methods are not much different. However, in the crushing process, the samples cooled by −20 °C refrigerator and liquid nitrogen are more brittle and easy to crush. Considering the use of liquid nitrogen cooling, the cost is high. A temperature of −20 °C cooling is therefore selected.

#### 2.3.4. Mass Ratio

It can be seen from Figure 7 and Figure 8 that when the carrier is 1:1, 1:3, 1:4, and 1:5, the dissolution in 5 min has reached about 80%. The rapid release of drugs in the early stage is easy to cause sudden release, which has an impact on the safety of drugs. Therefore, the drug loading ratio is 1:2.

#### 2.3.5. Screening Barrel Temperature

In general, the higher the temperature, the higher the heat energy provided, which is more conducive to the combination of drugs and carriers. However, it should also be considered that high temperature will degrade the drug and carrier. It can be seen from Figure 9, Figure 10 and Figure 11 that the lower the low temperature, the higher the dissolution of *Polygonum cuspidatum* extract. This may be related to the fact that the extract of *Polygonum cuspidatum* is more stable with the carrier at lower temperature. This experiment has also tried to set the barrel temperature to 130 °C, but it can not be extruded because the torque exceeds 75%. Therefore, the barrel temperature is determined to be 140 °C.

#### 2.3.6. Screw Speed

The screw speed affects the shear force and the residence time of materials in the barrel. When the screw speed is too slow, the retention time of materials in the barrel becomes longer, which can more fully absorb heat energy, but also lose part of the mechanical energy under the action of shear force. At the same time, for heat-resistant materials, too long residence time will lead to degradation. If the screw speed is too fast, the drugs may not be fully melted and mixed, but the mechanical energy is greater. It can be seen from Figure 12 that the rotational speed is 40 rpm and the cumulative dissolution is the highest. Therefore, the screw speed is determined to be 40 rpm.

### 2.4. Orthogonal Array Experimental Design

As can be seen from Table 2, taking the dissolution rate of the phosphate buffer at 6.8 pH for 15 min as the quantitative index, range R and analysis of variance showed that the order of various factors affecting the cumulative dissolution of resveratrol was C > A > B, that is, the primary and secondary factors affecting the cumulative dissolution of resveratrol were drug loading ratio, barrel temperature, and screw speed. All factors have influence, but the gap is not very large. The best process combination is A1B1C2, that is, the barrel temperature is 140 °C, the screw speed is 40 rpm/min, and the ratio of Polygonum cuspidatum extract to carrier hpmcas is 1:2. In addition, in our other experiments, the best process combination of RES-SD is that the barrel temperature is 140 °C, the screw speed is 80 rpm/min, and the content of RES in RES-SD is the same as that in PCE-SD. The attendant substances in PCE affected the screw speed in the optimal preparation process of SD prepared by HME.

### 2.5. Optimal Prescription Process Validation

As can be seen from Figure 13, the prepared solid dispersions of *Polygonum cuspidatum* extract were able to release completely at 45 min with significantly higher dissolution compared to the homemade *Polygonum cuspidatum* extract and PM. As can be seen from Figure 14, the dissolution difference between the three batches of solid dispersions prepared was small, which proves that this preparation process is stable and feasible.

### 2.6. Characterization of the PCE-SD

#### 2.6.1. FTIR Assay

The spectra were subtracted from the water vapor spectra. It can be seen from Figure 15 that on the curve of *Polygonum cuspidatum* extract, there are phenol hydroxyl stretching vibration bands at 3310 cm^−1^, and C = C stretching vibration of the benzene ring at 1590 cm^−1^ and 1511 cm^−1^. In the physical mixture, the bands at 3310 still exist, which proves that there is no chemical reaction between the excipient and the extract of Polygonum cuspidatum. However, in the solid dispersion, the peak corresponding to the phenolic hydroxyl group is no longer sharp and moves to low frequency. It is speculated that the excipients in the solid dispersion form hydrogen bond interaction with resveratrol.

#### 2.6.2. SEM

It can be seen from Figure 3, Figure 4, Figure 5, Figure 6, Figure 7, Figure 8, Figure 9, Figure 10, Figure 11, Figure 12, Figure 13, Figure 14, Figure 15 and Figure 16 that resveratrol in *Polygonum cuspidatum* extract exists in the form of block crystal, the micro morphology of carrier hpmcas is irregular loose porous block, P188 is smooth ball, and there are round holes in the section. The physical mixture includes resveratrol in the form of massive crystals, and carriers hpmcas and P188 are in an irregular form. No massive crystals are found in the solid dispersion, which is dense and irregularly arranged. It is speculated that resveratrol exists in the carrier in amorphous or molecular state.

#### 2.6.3. XRD

As shown in Figure 17, the X-ray diffraction patterns of each sample are shown, in which the PCE shows several characteristic peaks of resveratrol, such as 6.42°, 16.22°, 19.04°, 22.16°, 23.44°, 28.16°, etc. This shows that resveratrol exists in crystal structure in the extract of Polygonum cuspidatum. The carrier hpmcas has no obvious crystal peak, which corresponds to the DSC curve. The two characteristic diffraction peaks of poloxamer 188 (P188) are 19.60° and 23.94°, which proves that P188 has a crystal structure. The characteristic superposition peak of RES and carrier appeared in the physical mixture of Polygonum cuspidatum extract (PCE-PM), and the intensity decreased, indicating that there was no chemical interaction between RES and carrier in PCE. On the curve of PCE-SD, there is no obvious RES diffraction characteristic peak, which proves that there is no crystal structure RES in the PCE-SD.

### 2.7. Saturated Solubility

Table 3 shows that the saturated solubility of resveratrol in the solid dispersion of *Polygonum cuspidatum* extract is 2.78 times higher than that in *Polygonum cuspidatum* extract. This shows that the solid dispersion can effectively improve the dissolution rate of insoluble resveratrol. The reason may be that resveratrol exists in amorphous form in the solid dispersion, which increases the specific surface area during dissolution. In addition, the surfactant P188 is added to the prescription, which can also achieve the solubilization effect. The organic acid in the attendant substances may act as a solubilizing agent, making the saturated solubility of PCE higher than RES [30].

### 2.8. Stability Experiment

It can be seen from Table 4 that in the environment of 25 °C and 90% ± 5% RH, the moisture absorption rate of *Polygonum cuspidatum* extract and solid dispersion of *Polygonum cuspidatum* extract decreased after the fifth day, but both of them were greater than 5% as specified in Chinese Pharmacopoeia 2020. Therefore, the moisture absorption was further investigated in the environment with lower relative humidity. It can be seen from Table 5 that the hygroscopicity of PCE-SD can meet the requirements of the Pharmacopoeia when the environmental relative humidity is 70% ± 5%. While in our other experiments, the hygroscopicity of RES-SD in the environment of 25 °C and 90% ± 5% RH has already met the requirements of the Pharmacopoeia. This means the attendant substances in PCE affected hygroscopicity of SD in stability. The saccharides and tannins in PCE may be the reasons for the stronger hygroscopicity of PCE and PCE-SD [31].

It can be seen from Table 6 that the prepared PCE-SD improves the light sensitivity of RES. It can be seen from Table 7 and Table 8 that PCE-SD is relatively stable under high temperature and strong light environments.

### 2.9. Pharmacokinetic Study

According to the Figure 18 and pharmacokinetic parameters in Table 9, after intragastric administration, the Cmax of PCE group, PCE-PM group, and PCE-SD group were 761.161, 831.46 and 946.048μg·L^−1^, respectively, and AUC0-t were 111,471.22 ± 11.4, 146,598.478 ± 5.6 and 160,458.968 ± 15.7 μg·L^−1^, respectively. The bioavailability of PCE-SD group was 1.44 times higher than that of PCE group, and was slightly better than RES-SD in our other experiments. This means the attendant substances in PCE improve bioavailability of SD. The organic acid in the attendant substances may act as a solubilizing agent, making the bioavailability of PCE-SD higher than RES-SD [30].

## 3. Materials and Methods

### 3.1. Materials

The reference substances of Naringin and RES were purchased from Yuanye Biotechnology Co., Ltd. (Shanghai, China), and *Polygonum Cuspidatum* was obtained from Kemel Technology Co., Ltd. (Nanjing, Jiangsu, China). The raw RES was bought from Aladdin Biochemical Technology Co., Ltd. (Shanghai, China). Hydroxypropyl acetate methylcellulose succinate (HPMCAS) was received as a gift sample from Shin-Etsu Chemical Co., Ltd. (Tokyo, Japan), and Polosham 188 (P188) was received as a gift sample from Yunhong chemical preparation auxiliary material technology Co., Ltd. (Shanghai, China). All other chemicals and reagents used in this study were of analytical grade and obtained from Sinopharm Chemical Reagent Co., Ltd. (Shanghai, China).

SPF grade Sprague Dawley male rats, weighing 200–220 g, were purchased from Shanghai jiesijie experimental animal Co., Ltd. The animal certificate number is 20180004062158, and the animal license is scxk (Shanghai) 2018-0005. The test was approved by the animal ethics committee of Nanjing University of traditional Chinese medicine (Approval number 012009026349).

### 3.2. HPLC Analysis

The analysis of the drug was performed using a Waters2695 high-performance liquid chromatograph (Waters Corporation, Milford, MA, USA) with a UV detector. Using Ultimate Plus C18 (4.6 mm × 250 mm, 5 μm) chromatographic column, the solvent system used was 0.1% phosphoric acid: Acetonitrile (65:35) with a 1.0 mL/min flow rate. The column temperature was 30 °C, the injection volume was 10.0 μL, and the UV detection wavelength was 306 nm.

### 3.3. Preparation of Polygonum Cuspidatum Extract (PCE)

The root powder of *Polygonum cuspidatum* (50 g) was extracted twice by adding 10 times the amount of 80% ethanol in a water bath refluxing extraction at 60 °C for 1 h each time. The extracts were collected, filtered, and then concentrated by using a rotary vacuum evaporator (RE-52A Rotary evaporator; Yarong biochemical instrument factory, Shanghai, China) until all the possible alcohol was evaporated. Then added 1 times the amount of 30% ethanol with pH = 2, left for 12 h and then filtered to obtain the filtrate. After that, the filtered solution was concentrated with a rotary vacuum evaporator till all the possible alcohol was evaporated. The pH of the concentrated solution was adjusted to 7. Then, the concentrated solution was extracted with an equal volume of water saturated ethyl formate three times. The extract solutions obtained were combined and evaporated to dryness by a rotary evaporator under vacuum at 65 °C. The yield was calculated dividing the mass of recovered dry extract (mr) by the initial mass of P. cuspidatum powder (mi) (see Equation (1)), and the content of resveratrol was detected by HPLC.
Yield (%) = (mr/mi) × 100%(1)

### 3.4. Preliminary Qualitative Study on Main Attendant Substances in PCE

A preliminary qualitative study was carried out on the accompanying substances in PCE, using each component to examine the reaction. For example, the saccharides were detected by Fehling’s solution, and the amino acid was detected by Ninhydrin Test.

### 3.5. Preparation of PCE Solid Dispersions (PCE-SDs) and PCE Physical Mixtures (PCE-PMs)

#### 3.5.1. Hot-Melt Extrusion (HME)

Prior to HME, the materials were mixed using a vortex mixer (Vortex kylin-bell5 vortex oscillator, Baden Medical Co., Ltd., Nanjing, China) for 5 min to ensure homogeneous blending. The blended materials were manually added into the feeding port of the hot melt extruder and extruded on a 16 mm rotating twin-screw extruder (ThermoFisher Scientific Co., Waltham, MA, USA). Finally, the extrudate was placed in the freezer (BCD-255WDCI freezer, Haier, Qingdao, China) at −20 °C for more than 2 h, and crushed by a comminuting mill (Fw100 high speed universal crusher, Taist Instrument Co., Ltd., Tianjin, China), sieved (80 mesh) and stored in a dryer with discolored silica gel.

#### 3.5.2. Physical Mixture

The physical mixture of PCE-HPMCAS was prepared by vortexing it for 5 min after passing the materials through a 80 mesh sieve for use in the following experiments.

### 3.6. Evaluation Parameters

#### 3.6.1. DSC (Differential Scanning Calorimetry) Study

DSC (NETZSCH 200 F3 thermo gravimetrical analyzer, NETZSCH group, Selb, Germany) was utilized to check the physical state of RES (amorphous/crystalline) in the physical mixtures and in the extrudates after hot melt extrusion, as it can significantly affect the dissolution rate. A total of 5–10 mg of samples were accurately weighed and placed in an aluminum pan. The empty pans were sealed similarly, and were used as a reference. The heating range used was from 30 to 300 °C at a heating rate of 10 °C/min under nitrogen environment. Data analysis was performed using Pyris thermal analysis software (PerkinElmer, Massachusetts, USA).

#### 3.6.2. In Vitro Dissolution Studies

Dissolution studies were conducted using a USP II paddle method (75 rpm, 37 °C, and 900 mL of dissolution medium) with a dissolution test station (ZRS-8GD intelligent dissolution tester, Tianda Tianfa Company, Tianjin, China). An amount of SD powder equivalent to 40 mg of TEL was exposed for 45 min to pH 6.8 phosphate buffer (n = 3). A 2 mL sample was withdrawn from the dissolution medium at predetermined intervals (5, 10, 15, 20, 25, 30, and 45 min), and then the drug concentration was determined using HPLC; additionally, 2 mL of fresh medium was added to maintain a constant dissolution volume.

Phosphate buffer pH 6.8 was chosen as the in vitro dissolution medium because the extracts were most soluble in a buffered salt solution at pH 6.8 and considering the absorption of the drug in the intestinal fluid.

### 3.7. Univariate Analysis of Solid Dispersion of Polygonum Cuspidatum Extract

In this paper, the effects of the above factors on the content and release degree of PCE solid dispersion were investigated by Univariate analysis screening in terms of carrier type, release regulator type, cooling temperature, mass ratio of drug to carrier, preparation temperature, and screw speed. Extrudates were cooled at room temperature (25 °C), −20 °C refrigerator cooling, and liquid nitrogen cooling (−196 °C). The specific test was selected according to the evaluation index under “3.6”. The best prescription and preparation processes were obtained. The detailed results are shown in Table 10.

### 3.8. Orthogonal Array Experimental Design to Obtainoptimal Prescription of PCE-SD

According to single factor experiments, there were three significant factors in the dissolution of RES. Therefore, the barrel temperature (factor A), the ratio of drug and carrier (factor B), and the screw speed (factor C) were chosen as variables. The cumulative dissolution of resveratrol of 15min was selected as the dependent variable. The experiment used an L9(3)3 orthogonal test design. Range analysis was performed to reflect the optimal reaction conditions and their magnitudes. Each row of the orthogonal array represents a run, which is a specific set of tested factor levels. The run order of the trials was randomized to avoid the influence of unexplained variability in the observed response caused by systematic bias. Statistical significance was set as a *p*-value below 0.05 (* *p* < 0.05). The experiment was repeated under the optimal conditions three times in order to verify the data.

### 3.9. Optimal Prescription Process Validation

Based on the results of single-factor and orthogonal tests, the optimal prescription and process were determined. The extracts were weighed and mixed with excipients in the prescribed amount, then added into the hot melt extrusion, and the extrudate was cooled and crushed in the refrigerator at −20 °C, and the 80 mesh sieved sample, i.e., PCE-SD, was kept in a desiccator and stored away from light. The release degree test was performed on API, PCE-PM and PCE-SD. In addition, PCE-SD was prepared in three parallel batches, and the solid dispersions were examined according to the degree of dissolution.

### 3.10. Characterization of the Extrudate

#### 3.10.1. PXRD Study (Powder X-ray Diffraction)

X-ray diffraction was performed to assess the crystallinity of PCE, HPMCAS, P188, the physical mixtures, and the extrudates by Rigaku X-ray system (D/MAX-2500PC, Rigaku Corporation, Tokyo, Japan). The following experimental parameters were applied in all cases: CuKα radiation, operating at 40 kV and 40 mA, scanned over 2θ range of 5–40°, step width of 0.02°/S at a scanning speed of 2°/min.

#### 3.10.2. FTIR Study

FITR was used to investigate the drug-polymer interaction. The samples were prepared by mixing with potassium bromide (KBr) in a mortar and then pressing with a high-pressure clamp. The sample is tested by infrared spectroscopy (Thermo Nicolet, Madison, WI, USA) in the range of 400–4000 cm^−1^ and the resolution is 4 cm^−1^.

#### 3.10.3. Scanning Electron Microscopy

SEM was carried out to observe the micro state of resveratrol in excipients. The morphology and surface characteristics of PCE, HPMCAS, P188, PM, and SD were studied by SEM. The samples were placed on the substrate, and a layer of gold–palladium (Au/Pd) was sputter-coated as a conductive layer. Then, the samples were scanned with S-4800 SEM (Hitachi Limited, Tokyo, Japan) at the accelerating voltage of 30 kV to explore the changes in the surface characteristics of RES before and after formulating.

#### 3.10.4. Saturation Solubility

The saturation solubility of PCE and the best prepared formulations were measured by the shake flask method. The excess amount of material was added to the 6.8 phosphate buffer in a 50 mL centrifugal tube, and was kept in a shaking incubator (Hy-45 gas bath thermostatic biological shaker, Jintan Jincheng Guosheng experimental instrument factory, Nanjing, China) for 24 h at 37 °C. After 24 h, the 1 mL aliquots were removed and passed through 0.45 μm filter paper in order to remove the insoluble particles. The dissolved content in the aliquot was measured via HPLC at 306 nm.

#### 3.10.5. Stability Experiment

Stability of dosage forms was mainly affected by several factors, such as temperature, light, humidity, and oxygen content [32]. In this article, SDs are stored under high temperature, high humidity, and strong light conditions, and their content and crystallinity are investigated. In influencing factor tests, PCE-SD with a thickness of no more than 5 mm was placed in a flat weighing bottle, and then placed in a constant-temperature and humidity airtight dryer. Within 10 days, the content and the moisture absorption rate of RES was determined. In long-term retention test, PCE-SD was tightly enclosed in a 50 mL plug tube and stored at conditions (25 ± 2 °C, RH 65% ± 5%). The crystallinity of RES were checked at 1, 2, and 3 months.

### 3.11. Pharmacokinetics

In vivo pharmacokinetics was evaluated using male Sprague Dawley rats (200 ± 20 g). Eighteen Sprague Dawley rats were randomly divided into three groups of six rats each, fasted and drinking freely for 12 h before drug administration. The first group was gavaged with tiger wort extract suspension (PCE), the second group was gavaged with tiger wort extract mixed with physical mixture of excipients (PCE-PM), and the third group was gavaged with tiger wort extract solid dispersion suspension (PCE-SD), and the dose administered to each group was 50 mg RES/kg. At 5, 15, 30, 45, 60, 90, 120, 240, 360, 480, 600, 720, and 1440 min after intragastric administration, 1500 μL of blood in rat eyes was collected into heparin tubes and immediately centrifuged at 5000× *g* for 5 min to isolate the plasma. Approximately 200 μL of each plasma sample were kept at −20 °C until further analysis.

The RES concentration in the plasma was analyzed by HPLC. Briefly, 100 μL collected plasma samples were put with 50 μL naringin internal standard solution and 300 μL methanol in 1.5 mL microtubes, and the tubes were vortexed violently for 10 min and centrifuged at 13,000 R min^−1^ for 10 min. After that, the supernatant was tested in HPLC. Using Ultimate Plus C18 (4.6 mm × 250 mm, 5 μm) chromatographic column, the solvent system used was 0.1% phosphoric acid: Acetonitrile (65:35) with a 1.0 mL/min flow rate. The column temperature was 30 °C, the injection volume was 10.0 μL, and the UV detection wavelength was 306 nm.

## 4. Conclusions

PCE possesses attractive activities, but it has limited oral bioavailability because of its low aqueous solubility. In this study, PCE-SD was prepared and optimized using an orthogonal array experimental design. Analyses indicated that PCE was dispersed in the optimal PCE-SD formulation in the form amorphous molecularly dispersed particles; as a result, the optimal SD effectively enhanced the solubility and dissolution behavior of PCE. Furthermore, pharmacokinetic studies in rats indicated that the SD significantly improved the oral bioavailability of PCE. Therefore, a PCE-SD prepared with HPMCAS and P188 may be a promising approach to enhance the solubility, dissolution, and bioavailability of poorly soluble PCE. In addition, attendant substances in Polygonum cuspidatum extract do have effects in the preparation process, stability, and bioavailability of PCE-SD.

## Figures and Tables

**Figure 1 molecules-28-00737-f001:**
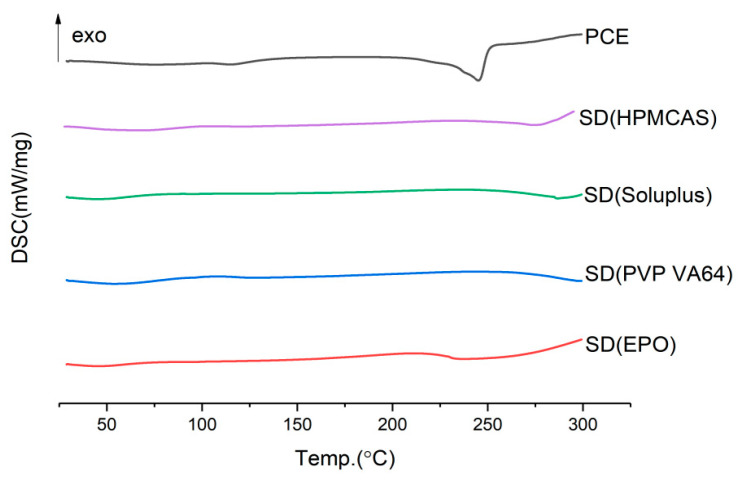
DSC diagram of solid dispersion prepared by different carrier.

**Figure 2 molecules-28-00737-f002:**
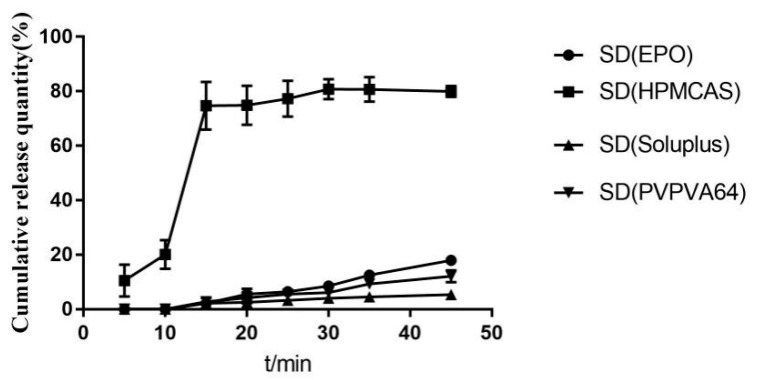
Dissolution curves of solid dispersion prepared by different carrier.

**Figure 3 molecules-28-00737-f003:**
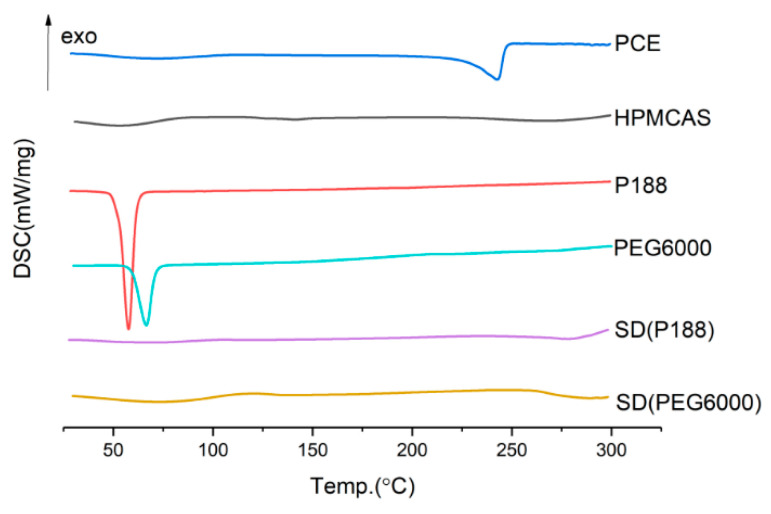
DSC diagram of solid dispersion prepared by different plasticizer.

**Figure 4 molecules-28-00737-f004:**
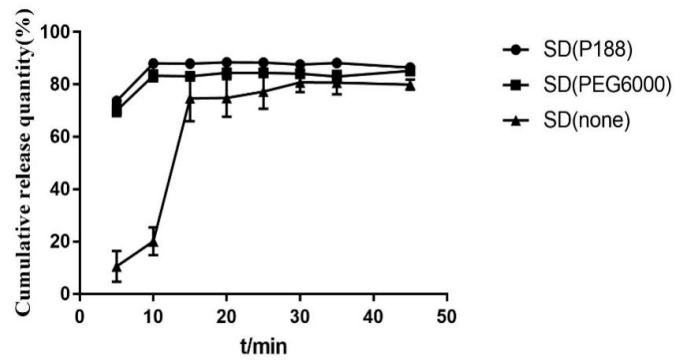
Dissolution curves of solid dispersion prepared by different plasticizer.

**Figure 5 molecules-28-00737-f005:**
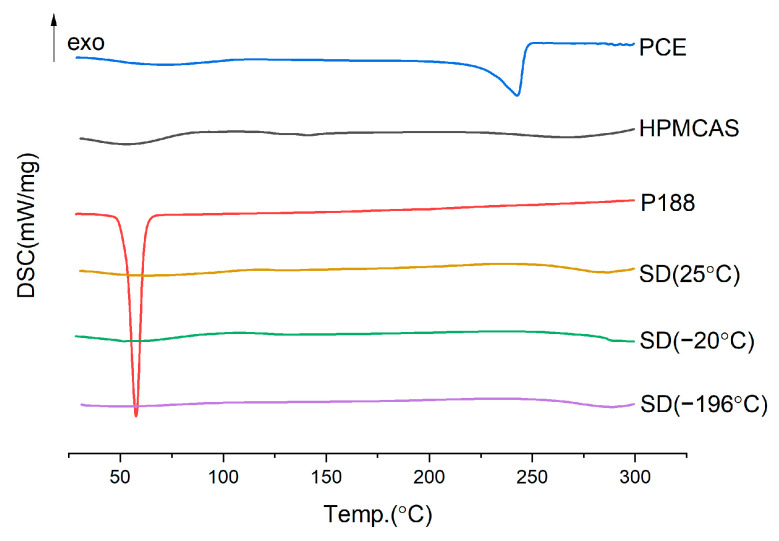
DSC diagram of solid dispersion prepared by different cooling temperature.

**Figure 6 molecules-28-00737-f006:**
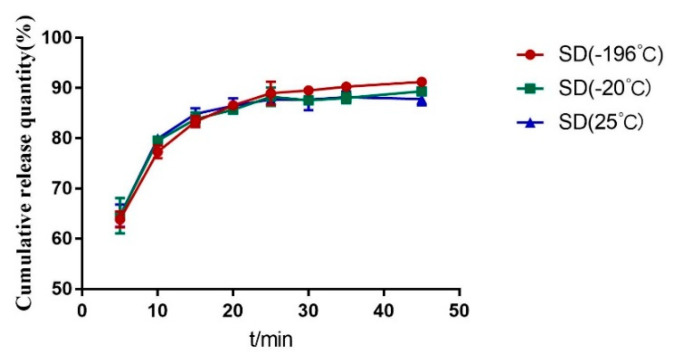
Dissolution curves of solid dispersion prepared by different cooling temperature.

**Figure 7 molecules-28-00737-f007:**
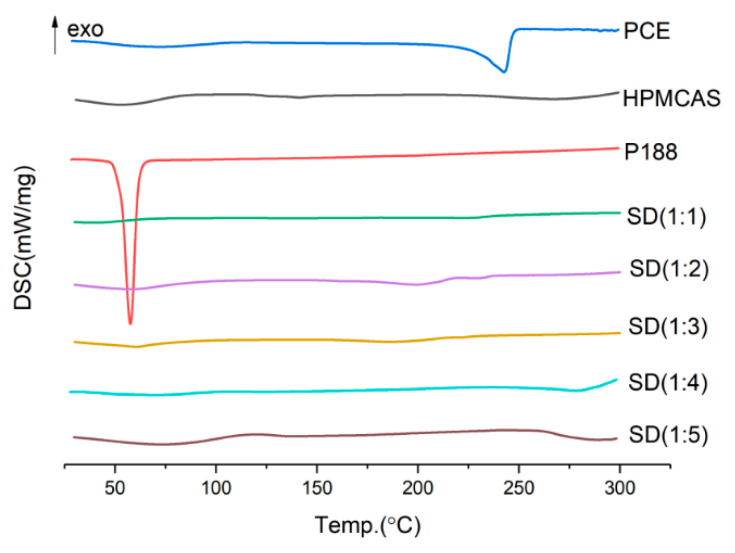
DSC diagram of solid dispersion prepared by different mass ratio.

**Figure 8 molecules-28-00737-f008:**
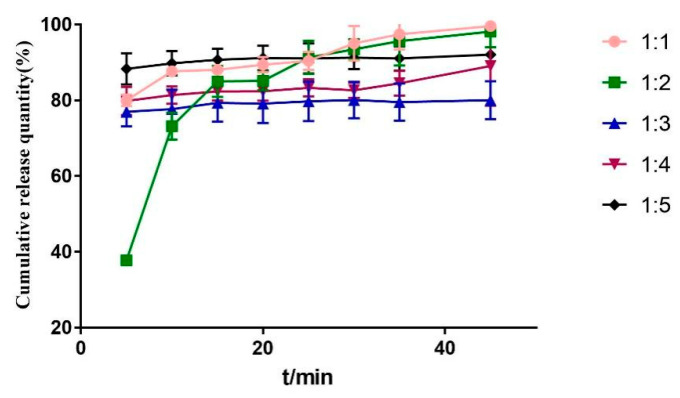
Dissolution curves of solid dispersion prepared by different mass ratio.

**Figure 9 molecules-28-00737-f009:**
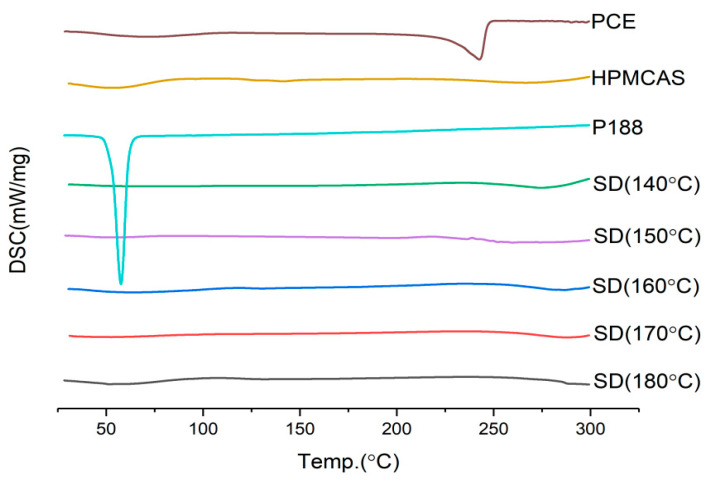
DSC diagram of solid dispersion prepared at different barrel temperature.

**Figure 10 molecules-28-00737-f010:**
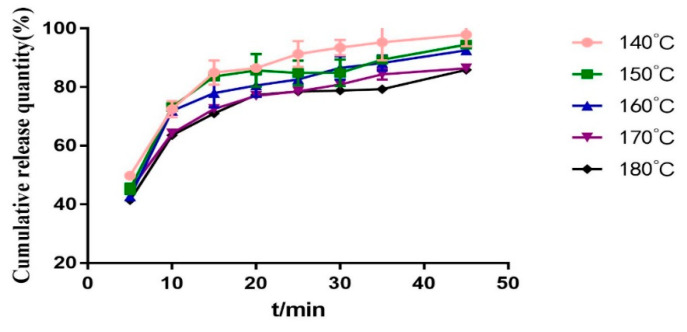
Dissolution curves of solid dispersions prepared at different barrel temperature.

**Figure 11 molecules-28-00737-f011:**
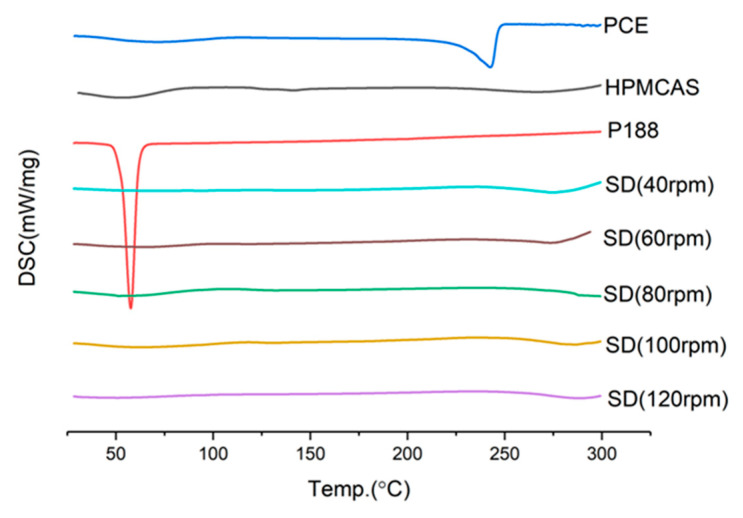
DSC diagram of solid dispersion prepared at different screw speeds.

**Figure 12 molecules-28-00737-f012:**
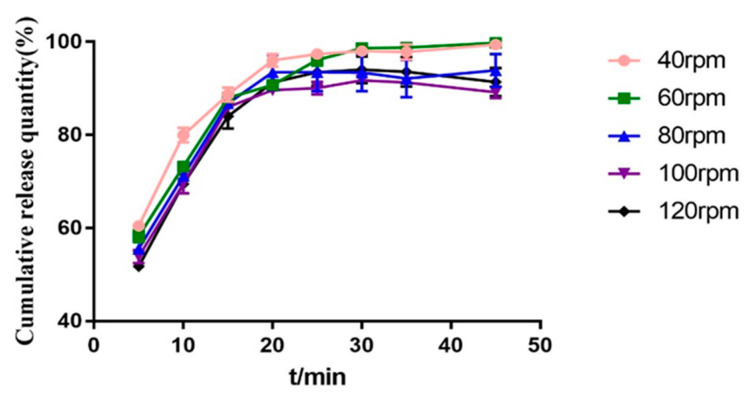
Dissolution curves of solid dispersions prepared at different screw speeds.

**Figure 13 molecules-28-00737-f013:**
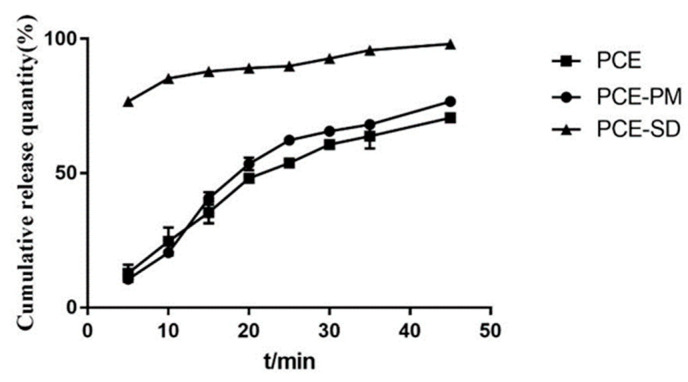
Dissolution curves of PCE, physical mixture and solid dispersion.

**Figure 14 molecules-28-00737-f014:**
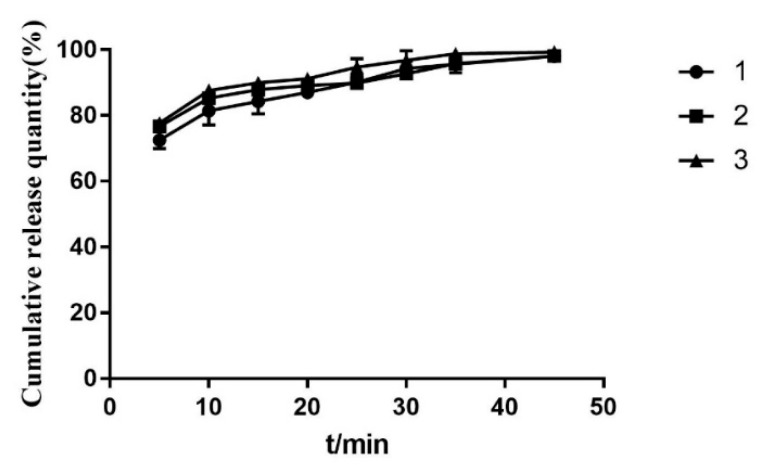
Dissolution curves of three batches of PCE solid dispersions.

**Figure 15 molecules-28-00737-f015:**
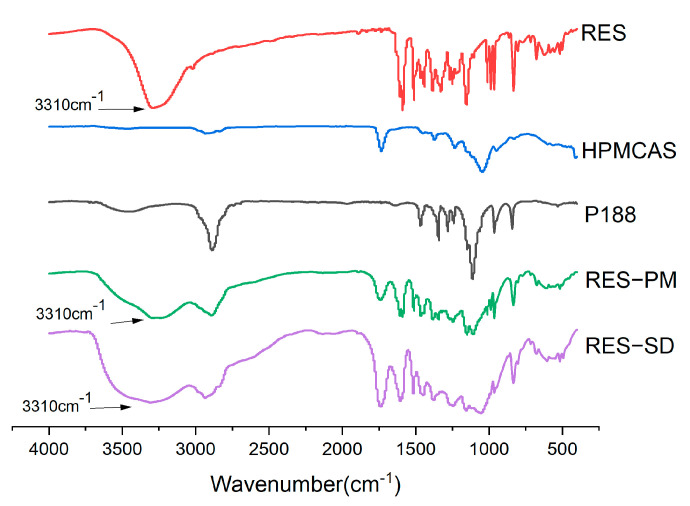
IR spectrum of PCE, carriers, PCE−PM and PCE−SD.

**Figure 16 molecules-28-00737-f016:**
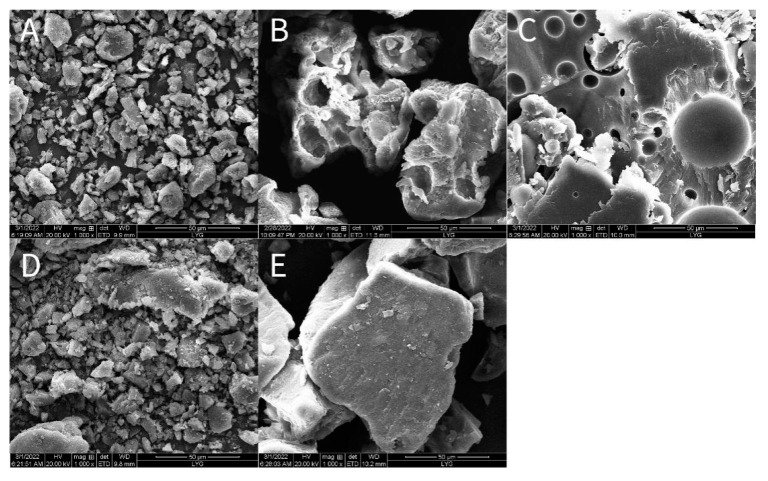
SEM microscopy of PCE (**A**), HPMCAS (**B**), P188 (**C**), PM (**D**) and PCE-SD (**E**).

**Figure 17 molecules-28-00737-f017:**
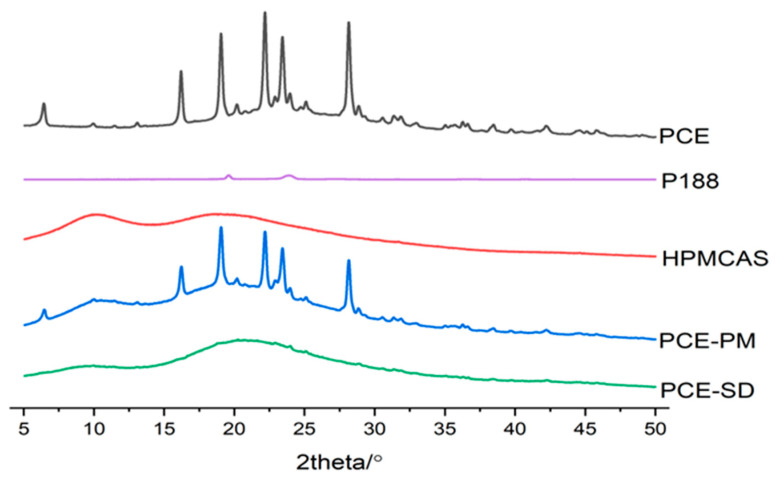
XRD spectrum of PCE, carriers, PM and PCE-SD.

**Figure 18 molecules-28-00737-f018:**
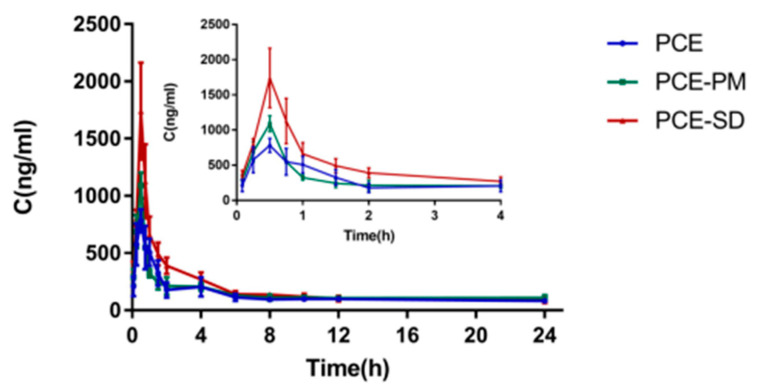
Plasma concentration-time curves in rats after oral administration of PCE, PCE-PM and PCE-SD.

**Table 1 molecules-28-00737-t001:** The chemical identification results on the composition of accompanying in PCE.

Inspection Items	Identification Reagent or Reaction Name	Experimental Result
Amino acid	Ninhydrin Test	(−)
Polypeptide, Protein	Biuret Reaction	(−)
Reducing sugars, Polysaccharides and Plycosides	Molisch Reaction	(+)
	Fehling’s Solution	(+)
Saponins	Foam Stability Test	(−)
Tannin	Liquor Ferri Trichloridi	(+)
	Sodium Chloride Gelatin Solution	(+)
Organic acid	pH Test Strips	(+)
	Bromocresol Green Test Solution	(+)
Alkaloid	Tetrapotassium Heptaiodobismuthate	(−)
Flavone	HCI-Mg Reaction Colorimetry	(−)
Anthraquinone	1% Sodium Hydroxide Solution	(+)

**Table 2 molecules-28-00737-t002:** Design and results of L9(3)3 orthogonal experiment.

Run	Factors			Responses
	Barrel Temperature (°C)	Screw Speed (rpm)	Mass Ratio	
1	140	40	1:1	90.15
2	140	60	1:2	92.26
3	140	80	1:3	93.18
4	150	40	1:2	90.87
5	150	60	1:3	85.79
6	150	80	1:1	82.08
7	160	40	1:3	90.69
8	160	60	1:1	81.55
9	160	80	1:2	87.88
K1	91.863	90.570	84.593	
K2	86.247	86.533	90.337	
K3	86.707	87.713	89.887	
Rj	5.616	4.037	5.744	

**Table 3 molecules-28-00737-t003:** Saturated solubility of PCE and PCE-SD.

	PCE	RES	PCE-SD
Saturated solubility (μg/mL)	46.75 ± 0.47%	44.34 ± 0.87%	130.06 ± 0.12%

**Table 4 molecules-28-00737-t004:** The content and moisture absorption under high humidity (saturated KNO3 solution, RH 90% ± 5%).

		PCE	PCE-SD
moisture absorption/%	0d	0	0
	5d	8.87	7.02
	10d	5.27	5.59
content/%	0d	50.24	99.83
	5d	44.83	97.29
	10d	39.88	92.36

**Table 5 molecules-28-00737-t005:** The content and moisture absorption under high humidity (saturated NaCl solution, RH 70% ± 5%).

		PCE	PCE-SD
moisture absorption/%	0d	0	0
	5d	4.93	4.45
	10d	3.27	3.07
content/%	0d	50.17	99.36
	5d	50.13	100.15
	10d	50.27	99.48

**Table 6 molecules-28-00737-t006:** The content of PCE and PCE-SD under strong light.

		PCE	PCE-SD
content/%	0d	50.24	100.05
	5d	44.19	98.05
	10d	36.36	99.52

**Table 7 molecules-28-00737-t007:** The content of PCE and PCE-SD under high temperature conditions.

		PCE	PCE-SD
content/%	0d	50.35	99.92
	5d	50.04	99.83
	10d	50.28	100.25

**Table 8 molecules-28-00737-t008:** The content of PCE and PCE-SD uinlong-term retention test.

		PCE	PCE-SD
content/%	30d	50.19	99.80
	60d	51.13	98.05
	90d	50.44	99.59

**Table 9 molecules-28-00737-t009:** Pharmacokinetic parameters of PCE, PCE-PM and PCE-SD in rats after oral administration.

Parameters	PCE	PCE-PM	PCE-SD
AUC0-t (μg/L·min)	111,471.22 ± 11.4	146,598.478 ± 5.6	160,458.968 ± 15.7
AUC0-∞ (μg/L·min)	496,970.649 ± 36.3	548,742.124 ± 59.1	283,435.733 ± 39.7
t1/2 z (min)	375.809 ± 44.8	226.132 ± 70.2	252.522 ± 79.1
Tmax (min)	21 ± 39.1	21.22 ± 11.0	19 ± 60.9
Cmax (μg/L)	761.161 ± 25.8	831.46 ± 18.7	946.048 ± 8.1

**Table 10 molecules-28-00737-t010:** Univariate analysis of solid dispersion of Polygonum cuspidatum extract.

No.	Carrier	Plasticizer	Cooling Temperature	Mass Ratio	Barrel Temperature (°C)	Screw Speed (rpm)
1	EPO	none	−20 °C	1:4	180	80
2	PVP VA64	none	−20 °C	1:4	180	80
3	Soluplus	none	−20 °C	1:4	180	80
4	HPMCAS	none	−20 °C	1:4	180	80
5	HPMCAS	P188	−20 °C	1:4	180	80
6	HPMCAS	PEG6000	−20 °C	1:4	180	80
7	HPMCAS	P188	25 °C	1:4	180	80
8	HPMCAS	P188	−20 °C	1:4	180	80
9	HPMCAS	P188	−196 °C	1:4	180	80
10	HPMCAS	P188	−20 °C	1:1	180	80
11	HPMCAS	P188	−20 °C	1:2	180	80
12	HPMCAS	P188	−20 °C	1:3	180	80
13	HPMCAS	P188	−20 °C	1:4	180	80
14	HPMCAS	P188	−20 °C	1:5	180	80
15	HPMCAS	P188	−20 °C	1:2	140	80
16	HPMCAS	P188	−20 °C	1:2	150	80
17	HPMCAS	P188	−20 °C	1:2	160	80
18	HPMCAS	P188	−20 °C	1:2	170	40
19	HPMCAS	P188	−20 °C	1:2	180	60
20	HPMCAS	P188	−20 °C	1:2	140	40
21	HPMCAS	P188	−20 °C	1:2	140	60
22	HPMCAS	P188	−20 °C	1:2	140	80
23	HPMCAS	P188	−20 °C	1:2	140	100
24	HPMCAS	P188	−20 °C	1:2	140	120

## Data Availability

The data presented in this study are available on request from the corresponding author. The data are not publicly available due to privacy.

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
