# Peer review of "Preparation of Solid Dispersion of Polygonum Cuspidatum Extract by Hot Melt Extrusion to Enhance Oral Bioavailability of Resveratrol"

_molecules, 2023, doi:10.3390/molecules28020737_

Round 1

Reviewer 1 Report

In this manuscript, authors prepared a solid dispersion formulation of Polygonum Cuspidatum extract with intention to enhance the oral bioavailability of its active constituent resveratrol. The authors claimed that the bioavailability of is slightly better of solid dispersion in compared to its physical mixture. Overall the manuscript is average in importance and some points need to be address before its consideration in molecules.

1.       Line 30 Authors started the introduction section with abbreviation. Abbreviations need to recheck throughout the manuscript excluding abstract section.

2.        Line 39, what RSV stand for?

3.       Line 92, authors need to provide the details of animal ethical approval number with date.

4.       Section 2.2 .  Author did not mention  here the reference of HPLC method used for analysis. If this was the developed by authors itself, then its validation procedure need to be included in the manuscript.

5.       line 223. Kindly mention the exact group designed here with total number of animal used in this study.

Line 226. How the dose of RES was decided for this study? Provide the reference for this doses.

6.       Section 2.11. There are many discrepancies in HPLC method described here in compared to section 2.2 . The organic modifier methanol or acetonitrile and volume of injection 20 or 10 ul. Kindly check it thoroughly

7.       Figure 18 is not clearly showing plasma concentration at different time point with diferent formulations. Kindly shrink the time point vetween 720-1440 min or expand its X axis

8.       Auhtors need to do statistical calculation to show wheather increase in bioavailability is significant or not. Mere mentioning the 1.44. times is not proving that it is worthy or not.

9. Line 471. PK studies was performed in rat but here mentioned as mice. Authro needed to check whole manuscript for discrepancies in writing

Author Response

  1. Thank you for your good suggestion. I rechecked the abbreviations throughout the manuscript. Corrections have been made in the revised manuscript.
  2. Thank you for your good advice. RSV" was modified to "RES" in line 39.
  3. Thank you for your good suggestion. Line 93 adds the animal ethics approval number information.
  4. Thank you for your good suggestion. The HPLC analytical method validation procedure in Section 2.2 has been sent to you as an attachment. Considering the integrity of the article, no validation procedures were added to the review.
  5. Thank you for your good advice. The animal experiments section has been revised in line 224, and the doses of RES refer to the following two papers.

[1] Chimento A, De Amicis F, Sirianni R, Sinicropi MS, Puoci F, Casaburi I, Saturnino C, Pezzi V. Advances in Improving Oral Bioavailability and Beneficial Effects of Resveratrol. International Journal of Molecular Science 2019 March 19;20(6):1381. DOI: 10.3390/IJMS20061381.

[2] Spogli R, Bastianini M, Ragonese F, Iannitti RG, Monarca L, Bastioli F, Nakashidze I, Brecchia G, Menchetti L, Codini M, Arcuri C, Mancinelli L, Fioretti B. Resveratrol in dihydrogen oxidation Solid dispersion on magnesium (Resv@MDH) microparticles improves oral bioavailability. Nutrients. 2018 Dec 5;10(12):1925. DOI: 10.3390/NU10121925.

  1. Thank you for your good advice. Line 239 of the revised manuscript has been corrected.
  2. Thank you for your good advice. The plasma concentration-time plot has its x-axis expanded in Figure 18.
  3. Thank you for your good advice. PCE-SD did not significantly increase bioavailability.
  4. Thank you for your good advice. A correction has been made to line 475 of the revised manuscript.

Reviewer 2 Report

A well-written manuscript with some points that needs improvement. See the file attached.

Author Response

Thanks for your good advice.The correction has been already done in the revised manuscript

Round 2

Reviewer 1 Report

Authors addressed most of the concerns and now can be consider for publication in molecules